# Clinical Outcomes in Fibrolamellar Hepatocellular Carcinoma Treated with Immune Checkpoint Inhibitors

**DOI:** 10.3390/cancers14215347

**Published:** 2022-10-30

**Authors:** Krista Y. Chen, Aleksandra Popovic, David Hsiehchen, Marina Baretti, Paige Griffith, Ranjan Bista, Azarakhsh Baghdadi, Ihab R. Kamel, Sanford M. Simon, Rachael D. Migler, Mark Yarchoan

**Affiliations:** 1The Sidney Kimmel Comprehensive Cancer Center, The Johns Hopkins University School of Medicine, Baltimore, MD 21231, USA; 2Division of Hematology and Oncology, Department of Internal Medicine, University of Texas Southwestern Medical Center, Dallas, TX 75390, USA; 3Department of Pediatrics, Texas Tech University Health Sciences Center, El Paso, TX 79410, USA; 4Department of Radiology and Radiological Science, The Johns Hopkins University School of Medicine, Baltimore, MD 21205, USA; 5The Fibrolamellar Registry, New York, NY 10028, USA

**Keywords:** fibrolamellar hepatocellular carcinoma, liver cancer, immune checkpoint inhibitor, anti-PD-1 therapy

## Abstract

**Simple Summary:**

Fibrolamellar hepatocellular carcinoma (FLC) is a rare form of liver cancer that affects children and young adults. Although immune checkpoint inhibitors (ICIs) including anti-PD-1 and anti-CTLA-4 are becoming standard of care in various cancers, including other forms of liver cancer, few studies have examined the safety and efficacy of ICIs in FLC. This study represents the largest multicenter, retrospective cohort of FLC patients receiving ICIs alone and in combination with other drugs. Our results demonstrate that ICIs have modest clinical benefit in the treatment of FLC. Results of this analysis have important implications for the management of patients with FLC and will inform future treatment decisions.

**Abstract:**

Background: Fibrolamellar hepatocellular carcinoma (FLC) is a rare form of liver cancer primarily affecting children and young adults. Although considered a subset of hepatocellular carcinoma (HCC), FLC has unique molecular and pathologic characteristics, suggesting that it may require different treatment. Immune checkpoint inhibitors (ICIs) are used in the treatment of HCC, but efficacy and safety in FLC has not been characterized. Methods: We performed a multicenter retrospective analysis of patients with FLC to determine responses to ICI therapy. Response rates were assessed based on RECIST 1.1 criteria, and Kaplan–Meier statistics were used for progression-free survival (PFS) and overall survival (OS). Results: FLC tumors were characterized by low tumor mutational burden (TMB) and absent PD-L1 expression. We identified 19 patients who received ICIs, including 15 who received ICI therapy alone [programmed death receptor 1 (PD-1) inhibitor, +/− cytotoxic T lymphocyte antigen-4 (CTLA-4) inhibitor]. Objective tumor responses were observed in 3/19 patients (15.8%), including 2/15 patients (13.3%) who received ICIs alone, all partial responses. Median PFS and OS were 5.5 and 26.0 months, respectively. Grade 3–4 immune related adverse events were observed in 4/19 (21.1%) patients. Conclusions: ICI therapy has modest clinical activity in FLC, and novel therapeutic combinations are needed.

## 1. Introduction

Fibrolamellar hepatocellular carcinoma (FLC) is a rare and aggressive form of primary liver cancer. The signature genomic event is a gene fusion of the *DNAJB1-PRKACA*. The resulting DNAJB1-PRKACA fusion kinase is present in nearly all cases of FLC and its expression is sufficient to recapitulate FLC. Thus, it is a presumed oncogenic driver [1,2,3]. FLC primarily occurs in adolescents and young adults, usually in the absence of underlying liver disease or cirrhosis. Because of its prevalence in an otherwise healthy population in which clinical suspicion of cancer is low, diagnosis of FLC is often delayed [4,5]. For patients diagnosed with early stage disease, surgical resection or liver transplantation remain the only potentially curative treatment options and are commonly considered the greatest predictors of overall survival in FLC patients [6,7,8]. However, 40–50% of patients present with advanced or late stage disease at diagnosis [8,9].

These is no standard systemic therapy for patients with FLC, and the management of unresectable FLC has largely been extrapolated from the management of advanced hepatocellular carcinoma (HCC), despite marked biological differences between these cancers [10]. FLC patients are generally excluded from clinical trials of HCC, and there are limited prospective studies to guide therapeutic selection in the treatment of FLC. Although historically characterized as a tumor type with indolent biology, FLC is often resistant to systemic therapies developed for other tumor types, and the median overall survival for patients with advanced FLC in a modern retrospective series was 12 months [7]. Given the rareness of FLC and the limited availability of prospective clinical studies, retrospective cohort studies may provide important information to guide future treatment strategies.

Immune checkpoint inhibitors (ICIs) targeting programmed cell death protein 1 (PD-1), or its ligand (PD-L1), have transformed the treatment of many different cancers and are used in the standard management of HCC [11,12]. However, the effectiveness of ICI therapy in FLC is largely unknown. We conducted a multicenter retrospective cohort study of patients with FLC who were treated with anti-PD1 or anti-PDL1 therapies using data from the Johns Hopkins Liver Cancer Database, the UT Southwestern Medical Center Liver Cancer Database, and the Fibrolamellar Registry, an international registry of patients with FLC.

## 2. Materials and Methods

### 2.1. Study Design

This is a retrospective cohort study of patients with FLC, identified using the Johns Hopkins Liver Cancer Database, the UT Southwestern Medical Center Liver Cancer Database, and the Fibrolamellar Registry. Eligible patients were defined as those diagnosed with advanced stage FLC treated with anti-PD-1 or anti-PDL1 therapy, alone or in combination with other therapeutic agents. Only patients with a confirmed pathologic diagnosis of FLC by the institution where the patient received treatment, with no history of prior ICI treatment, and who had medical records and baseline on-treatment imaging available for analysis were included. Pathology was not centrally reviewed, and confirmation of the DNAJB1-PRKACA fusion kinase was not required for patients to be included in the study. Patients receiving ICI therapy as part of an ongoing interventional clinical trial were excluded, as were patients receiving immunotherapy as standard of care who had previously participated in a clinical trial of cancer immunotherapy.

Eligible patients identified through the Fibrolamellar Registry (http://fibroregistry.org accessed on 24 September 2022) made their medical records, CT scans, and MRI scans available to the Registry, and the study team received medical records and scans in a de-identified fashion. Manual chart reviews for all patients in our cohort collected the following information: demographics (age at diagnosis, gender); clinical history prior to ICI therapy (tumor stage at diagnosis, extrahepatic disease at diagnosis, treatments prior to ICI therapy including start and end dates and treatment response, tumor stage at ICI treatment initiation, extrahepatic disease at ICI treatment initiation); ICI treatment course (start date, end date, number of doses, time of therapy, response, ICI related toxicities); and vital status. All staging was defined in accordance with the Barcelona Clinic Liver Cancer Staging System (BCLC). We additionally examined molecular profiling of all FLC patients with genomic data available in the Johns Hopkins Liver Cancer Database.

### 2.2. Assessments and Analyses

CT and MRI scans were interpreted in accordance with Response Evaluation Criteria in Solid Tumors (RECIST 1.1) [13]. RECIST 1.1 reads from the Fibrolamellar Registry and from Johns Hopkins were confirmed by a liver radiologist (author IK). All patients who received at least one dose of ICI therapy and had a follow-up scan available for analysis were considered to be evaluable for response. Progression-free survival (PFS) and overall survival (OS) were calculated according to Kaplan–Meier methodology using R version 4.1.3. These were calculated from ICI start date until date of disease progression (PFS) and death (OS). If patients were still on therapy without progression, PFS was censored on date of last available scans. If patients were lost to follow-up, OS was censored on date last known to be alive.

## 3. Results

### 3.1. FLC Is Characterized by a Low Tumor Mutation Burden (TMB) and Low PD-L1 Positivity

To understand the immune landscape of FLC and potential for sensitivity to ICI therapy, we first examined TMB, a surrogate for tumor neoantigen quantity [14], among all patients in the Johns Hopkins Liver Cancer Database with FLC who had received molecular profiling as standard of care. Approximately half of the differences in the objective response rate across cancer types may be explained by the TMB, with higher TMBs correlating with higher benefit [15,16]. Consistent with many other pediatric cancer types [17], TMB was low in FLC, with a median TMB of 1.85 mut/MB (range 0–6 mut/MB) (Table 1). All 14/14 (100%) patients had a TMB of less than 10 mut/MB, which is a TMB threshold that is associated with tumor-agnostic ICI benefit [18].

Higher expression of PD-L1 assessed by immunohistochemistry (IHC) is also associated with clinical benefit from ICIs in some cancer types, and can provide independent information regarding immune sensitivity from TMB [19]. PD-L1 expression was negative (<1%) in 11/11 patients (100%) by IHC staining using archival specimens obtained prior to the initiation of therapy (Table 1). No patients had mismatch repair deficiency (MMRd). Thus, the immune profile of FLC is consistent with an immune resistant tumor type.

### 3.2. Clinical Characteristics

To determine the immune responsiveness of FLC to ICI therapy, we performed a retrospective analysis of patients with FLC in the Fibrolamellar Registry, the UT Southwestern Medical Center Liver Cancer Database, and the Johns Hopkins Liver Cancer Database who were treated with ICI therapy. A total of 19 patients met our eligibility criteria. Twelve (63.2%) of the patients were male, and 7 (36.8%) of the patients were female. Mean age of diagnosis was 22.9 years. Prior to initiation of ICI therapy, the majority of the patients had some form of local therapy including resection (13/19, 68.4%) or radiation (4/19, 21.1%). Most patients (15/19, 78.9%) received prior systemic therapy, with a median of 1 (range 0–8) prior lines of systemic therapy. By the time of ICI treatment initiation, 18 patients (94.7%) had metastatic stage FLC and 1 patient (5.3%) had liver-confined disease. Overall, the clinical characteristics were similar across record sources and databases. The only notable difference was that patients from Johns Hopkins and UT Southwestern were more likely to have late stage, metastatic disease at diagnosis (Table 2).

Immune checkpoint inhibitors (i.e., nivolumab monotherapy, pembrolizumab monotherapy, or nivolumab plus ipilimumab) were administered in 15 patients (78.9%). Of these, 9 received nivolumab (47.4%), 4 received pembrolizumab (21.1%), and 2 received nivolumab and ipilimumab in combination (10.5%). Among the 4 patients who received pembrolizumab, 2 of these patients also received cryoablation or other locoregional therapies to the liver concurrently but had non-ablated lesions that could be evaluated for systemic response.

Immune checkpoint inhibitors were administered in combination with other therapeutic agents in 4 patients. These included atezolizumab and bevacizumab (n = 1), nivolumab and 5FU and interferon alfa-2b (n = 1), nivolumab and regorafenib (n = 1), and nivolumab with multiple different agents in combination (gemcitabine, bevacizumab, capecitabine, interferon, and lenvatinib) (n = 1). At time of study endpoint, 5 patients were reported to still be on therapy. The median time on therapy was 5.1 months (range 0–36.5) (Table 3).

### 3.3. Efficacy

Among the 19 study patients, 3 patients (15.8%) experienced partial response, 4 (21.1%) experienced stable disease, and 12 (63.2%) experienced progressive disease as a best response, leading to an overall response rate (CR + PR) of 15.8% (Table 3). In the subset of patients receiving ICI therapy alone (anti-PD1 +/− CTLA4) (n = 15), the response rate was 2/15 (13.3%). The responses to ICI therapy were durable, lasting at least 8 months in all patients (Figure 1). Two patients had mixed responses to therapy, with reductions in target lesions but the appearance of new lesions, for an overall response of PD. Multiple patients had stable disease as a best response to therapy, although all of the patients had relatively indolent tumor growth both before and after initiation of ICI therapy, making it unclear whether the ICI therapy contributed to tumor control.

The treatment course of the subset of patients with FLC with objective responses to immunotherapy is summarized below. The first patient was a 29-year-old female diagnosed with metastatic disease who received sorafenib in first line with intolerance and was subsequently switched to nivolumab monotherapy. This patient developed multiple grade 3–4 immune-related adverse events and nivolumab therapy was discontinued after a single dose of treatment. However, the patient’s first interval scan demonstrated tumor regression that continued for almost a year despite receiving no further systemic therapy. Sum of target lesions was 17.05 cm at initial scan and decreased to 6.36 cm at 265 days as a best response. The partial response was attributed to be as a result of nivolumab therapy by the treating physician. The second patient was a 19-year-old male diagnosed with locally advanced disease who received sorafenib resulting in progressive disease and subsequent debulking surgery prior to treatment with 5FU + interferon alfa-2b + nivolumab. ICI treatment was initiated at metastatic stage cancer, and the patient had a partial response that was continuing at the time of analysis 8 months later. Sum of target lesions was 9.09 cm at initial scan and decreased to 4.62 cm at 270 days as a best response. The third patient was a 29-year-old male diagnosed with metastatic disease who received liver resection and cholecystectomy followed by sorafenib with progression at 5 months. He was subsequently treated with nivolumab with partial response lasting almost 1 year at the time of last analysis. Sum of target lesions was 18.79 cm at initial scan and decreased to 5.48 cm at 352 days as a best response.

The median progression-free survival from time of ICI therapy start date for patients receiving ICIs alone was 5.5 months (Figure 2). For all patients in the cohort, median overall survival from time of ICI therapy start date to date of death or last follow-up was 26.0 months. For the subset of patients who received ICI therapies alone without other concurrent therapy, the median progression free survival and overall survival were similar to the overall population (Figure 3). These data were also disaggregated to compare median progression-free survival and median overall survival between patients who received ICIs alone and ICIs in combination with other therapies (Appendix A)

### 3.4. Safety

There were no treatment-related deaths in any patients. Two out of nineteen patients (10.1%) reported discontinuation of ICI therapy due to adverse events (AEs). Eleven patients (57.9%) experienced any AE, and the average number of AEs experienced by a patient was 1.1. The most common all-grade toxicities during treatment were fatigue (21.1%), elevated liver function tests (LFTs, 21.1%), hypothyroidism (10.1%), and nausea (10.1%). Additionally, 4 patients (21.1%) experienced grade III/IV AEs (Table 4). These results were also disaggregated to compare AE data between patients who received ICIs alone and ICIs in combination with other therapeutic agents (Appendix A).

## 4. Discussion

There are limited prospective clinical trials to guide treatment selection in FLC. ICI therapies are used in the standard treatment of the two most common forms of primary liver cancer, hepatocellular carcinoma and cholangiocarcinoma, but the safety and efficacy of ICI therapy in FLC is unclear. To our knowledge, this is the largest retrospective cohort study of FLC treated with ICIs alone or in combination with other agents. Here, we show that ICIs do have single agent activity in FLC, with durable partial responses observed in 2/15 (13.3%) patients who received ICIs alone without any other concurrent agent and in 3/19 (15.8%) of our entire cohort, which included patients receiving ICIs concurrently with other therapies. The modest response rates to ICI therapy reported in this retrospective study are lower than response rates reported in prospective studies of anti-PD1 therapy in adults with hepatocellular carcinoma [20,21].

The ICI response rates observed in our study are in concordance with those previously reported in the literature for FLC. These include individual case reports, single-center case series, and a small prospective clinical trial of combination therapies that included ICIs. In one series, 3 patients at one institution were all shown to have progressive disease between 2 and 3 months of ICI therapy [22]. One report described a single patient who received combination therapy with nivolumab and ipilimumab and experienced a near-complete response [23]. There was also a recent interim report of neratinib alone or in various doublet or triplet combinations with everolimus and ICI therapy (pembrolizumab or nivolumab) in FLC. In this study, the response rate to the ICI-containing combination regimens was 1/7 (14.3%) [24]. Since neratinib had limited single agent activity, this objective response may reflect ICI activity. Collectively, these data suggest a modest benefit from ICI therapy in FLC. An ongoing multicenter prospective clinical trial of pembrolizumab in pediatric hepatocellular carcinoma (NCT04134559) including FLC will provide additional information about response rates to ICI therapy in this population.

Limited efficacy of ICI monotherapy described in this study may be due to intrinsic tumor characteristics of FLC such as low immunogenicity due to low TMB, negative PD-L1 status, two widely used biomarkers of ICI sensitivity, and a broadly immunosuppressive tumor microenvironment with upregulation of multiple coinhibitory molecules [25]. Additional studies are warranted to elucidate the immunological landscape in FLC and guide future treatment. Furthermore, due to the small number of patients in our study who received ICIs in combination with other therapies, additional studies are needed to evaluate whether ICIs may be effectively combined with other therapies to enhance clinical benefit.

There are a number of limitations to this study, including a relatively small sample size, lack of a control cohort, and possible selection biases inherent to the retrospective nature of the study. Although the Johns Hopkins and UT Southwestern cohorts represent all FLC patients treated with anti-PD1 therapy at these institutions over the study period, it is possible that patients with longer overall survival would be more likely to participate in the Fibrolamellar Registry, from which many of our cases were obtained. The relatively small sample size was anticipated, given the rarity of FLC and the recent introduction of ICIs in standard clinical management of HCC.

## 5. Conclusions

This study is the largest reported analysis of FLC patients receiving ICIs and further emphasizes the need for additional studies informing systemic treatment strategies for advanced FLC. We demonstrate that ICIs have modest clinical activity in the setting of FLC. Further investigation is warranted to interrogate the tumor immune microenvironment in responders and non-responders to immunotherapy in FLC and to develop biomarkers and identify novel, rational combinations to overcome barriers of effective antitumor immunity in FLC.

## Figures and Tables

**Figure 1 cancers-14-05347-f001:**
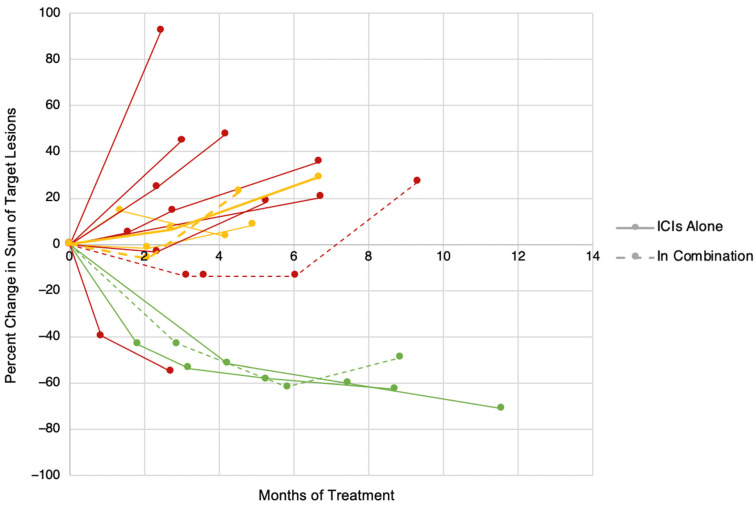
Percent change in sum of target lesions over time for patients with FLC treated with ICI therapy by RECIST 1.1 Criteria. Green = PR, Yellow = SD, Red = PD.

**Figure 2 cancers-14-05347-f002:**
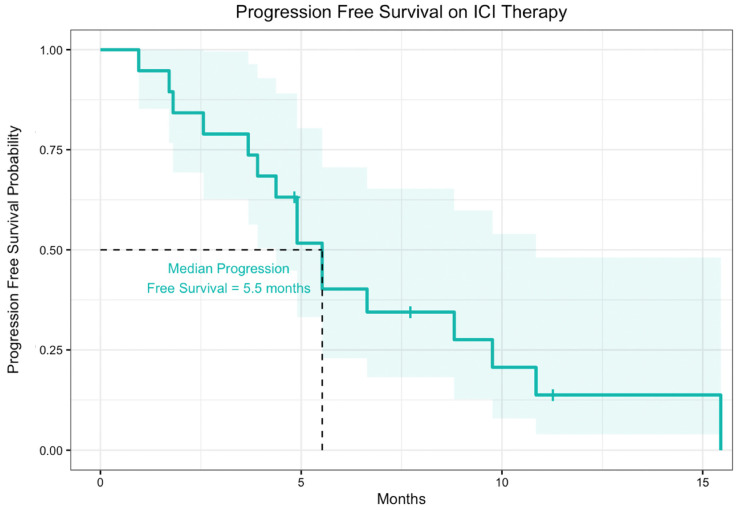
Progression free survival of 19 patients with FLC treated with ICI. Of 19 patients, 15 received ICIs alone and 4 received ICIs in combination with other therapies. The shaded region represents the 95% confidence interval.

**Figure 3 cancers-14-05347-f003:**
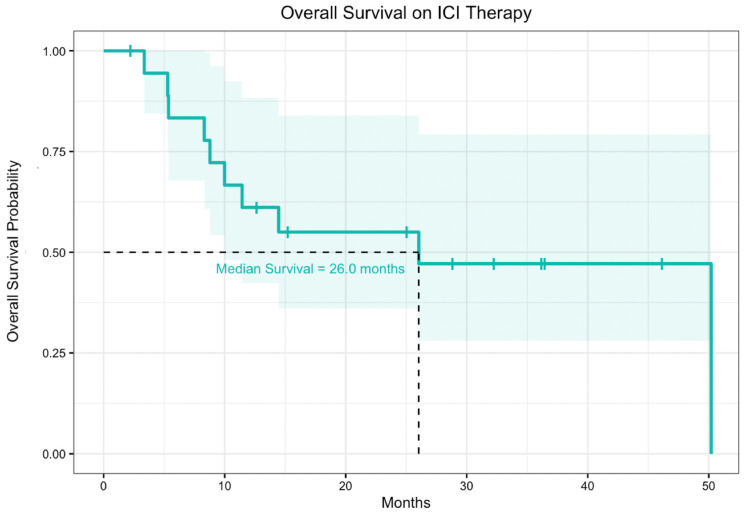
Overall survival of 19 patients with FLC treated with ICI therapy. Of 19 patients, 15 received ICIs alone and 4 received ICIs in combination with other therapies. The shaded region represents the 95% confidence interval.

**Table 1 cancers-14-05347-t001:** TMB and PD-L1 expression of all patients with FLC in the Johns Hopkins Liver Cancer Database with testing available for analysis.

PD-L1 by IHC	TMB (Mutations/MB)	Sequencing Provider
Negative (<1%)	2.6	Tempus
Negative (<1%)	4.2	Tempus
Negative (<1%)	1.1	Tempus
Negative (<1%)	6	Caris
Negative (<1%)	4	Caris
Negative (<1%)	6	Caris
N/A	1	Strata
Negative (<1%)	1	Caris
Negative (<1%)	1	Caris
N/A	0.5	Tempus
Negative (<1%)	4	Caris
Negative (<1%)	0	Tempus
N/A	0.5	Tempus
Negative (<1%)	4.7	Tempus

**Table 2 cancers-14-05347-t002:** FLC Patient Demographics and Clinical Characteristics.

Variable	Classification	Overall	Fibrolamellar Registry	Johns Hopkins & UT Southwestern
Number of Patients		19	11	8 (6 JH, 2 UTSW)
Age at Diagnosis (years)		Mean = 22.9 (SD = 6.1)	Mean = 20.7 (SD = 6.0)	Mean = 25.875 (SD = 5.1)
Gender	Male	12 (63.2%)	7 (63.6%)	5 (62.5%)
	Female	7 (36.8%)	4 (36.4%)	3 (37.5%)
FLC Stage at Diagnosis	BCLC A	3 (15.8%)	3 (27.3%)	0 (0%)
	BCLC B	4 (21.1%)	3 (27.3%)	1 (12.5%)
	BCLC C	12 (63.2%)	5 (45.5%)	7 (87.5%)
FLC Stage at ICI Treatment	BCLC A	0 (0%)	0 (0%)	0 (0%)
	BCLC B	1 (5.3%)	1 (9.1%)	0 (0%)
	BCLC C	18 (94.7%)	10 (90.9%)	8 (100%)
Number of Patients Receiving Various Treatments	Prior Systemic Treatment	15 (78.9%)	9 (81.8%)	6 (75.0%)
	Prior Surgery	13 (68.4%)	7 (63.6%)	6 (75.0%)
	Prior Local Radiation	4 (21.1%)	2 (18.2%)	2 (25.0%)
Number of Prior Systemic Treatments		Median = 1 (range 0–8)	Median = 2 (range 0–8)	Median = 1 (range 0–5)
	Prior Sorafenib	9 (47.3%)	9 (81.8%)	0 (0%)
Treatment Setting	Academic	18	10 (90.9%)	8 (100%)
	Community	1	1 (9.1%)	0 (0%)

**Table 3 cancers-14-05347-t003:** Clinical Characteristics of ICI Therapy for FLC patients.

Variable	Classification	N (%)
Immune Checkpoint Inhibitor	Nivolumab monotherapy	9 (47.4%)
	Nivolumab + 5FU + IFN	1 (5.9%)
	Nivolumab + regorafenib	1 (5.9%)
	Nivolumab + gemcitabine-based chemotherapy	1 (5.9%)
	Nivolumab + ipilimumab	2 (11.8%)
	Pembrolizumab monotherapy	4 (23.5%)
	Atezolizumab + bevacizumab	1 (5.3%)
Time on Therapy (months)		Median = 5.1 (range 0–36.5)
Any ICI Regimen	Progressive Disease	12 (63.2%)
	Stable Disease	4 (21.1%)
	Partial Response	3 (15.8%)
	Complete Response	0 (0%)
ICI Alone Subset (i.e., anti-PD1 +/− CTLA4)	Progressive Disease	10 (66.7%)
	Stable Disease	3 (20%)
	Partial Response	2 (13.3%)
	Complete Response	0 (0%)

**Table 4 cancers-14-05347-t004:** Adverse events in FLC patients due to ICI therapy.

**Number of patients who discontinued** **ICIs due to AE**	2 (10.1%)
Any grade III or IV AE	4 (21.1%)
Specific AE Data	
Elevated LFTs	4 (21.1%)
Fatigue	4 (21.1%)
Hypothyroidism	2 (10.1%)
Nausea	2 (10.1%)
Anaphylaxis	1 (5.3%)
Diarrhea	1 (5.3%)
Hyperthyroidism	1 (5.3%)
Pneumonitis	1 (5.3%)
Pruritis	1 (5.3%)
Pyrexia	1 (5.3%)
Rash	1 (5.3%)
Vomiting	1 (5.3%)
Average Number of AEs experienced by patients	1.1

## Data Availability

All data relevant to the study are included in the article or are available from the corresponding author on reasonable request.

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
