# Peer review of "Clinical Outcomes in Fibrolamellar Hepatocellular Carcinoma Treated with Immune Checkpoint Inhibitors"

_cancers, 2022, doi:10.3390/cancers14215347_

Round 1
Reviewer 1 Report
This is a valuable contribution from a leading investigator in the field of cancer immunotherapy. In this manuscript, the authors combine databanks from two leading cancer treatment centers with a unique patient data registry. Although the results are not overwhelmingly positive, they are a very valuable contribution to the care of patients with fibrolamellar carcinoma, where there is no current standard of care. There are some minor changes that could be made to the manuscript to increase its utility:
1. All patients with PR received prior sorafenib. If the information is available, it would useful to know what proportion of patients overall received prior sorafenib and whether prior (but not contemporaneous) anti-VEGF therapy may be associated with benefit.
2. The inclusion of a spider plot separated by ICI alone vs. combinations is valuable. Would recommend showing the ICI alone vs. ICI+non-ICI agents for OS/PFS curves and AE table. AEs may be increased in patients receiving combination therapy.
3. The regimen of gemcitabine, capecitabine, bevacizumab and interferon is unusual. Did this patient have a mixed response? Did they have multiple AEs?
4. The academic/patient advocacy collaboration here is a unique and valuable element of the report, but (as pointed out by the authors) could have some systematic differences between populations. Given the small size of the study, it may be difficult to contrast outcomes between the fibrolamellar registry and the academic sites, but comparing the demographics of each cohort could be valuable. If possible, the context in which these patients were treated (ie academic vs. community) would also be of potential interest.
Reviewer 2 Report
Dear authors,very interested article.Can you explain the rationale to use immunotherapy in patients with no predictirs for response(TMB,PDL-1,MSI)?Why do you use RECIST 1.1 but not IRECIST.?Ciuld yiu provide more details of responders with :course if disease,previous response to treatment,CBC with Ne/Ly ratio,size of primary tumour.What is about details about tha patients with SD?
Reviewer 3 Report
In the present manuscript by Krista Y. Chen and colleagues have attempted to demonstrate the benefits of immune checkpoint inhibitors (ICIs) used either as a stand-alone or in-combination therapy for rare liver cancer - fibrolamellar hepatocellular carcinoma (FLC). Using the available data from different centers, the authors have performed a retrospective analysis and tried to determine the effect of the use of ICIs as therapy in FLC.
This manuscript is well-presented, informative, and comprehensive in the analysis (given the resources and number of patients available), in demonstrating the activity of ICIs in FLC.
In the manuscript, adding one clarification can strengthen the importance of the manuscript:
1) In lines # 84 – 85, the authors have mentioned the eligibility of the patients considered in this study – which were either on anti-PD-1 or anti-PDL1 therapy. Secondly, in lines # 133 – 134, the authors mention IHC staining for PD-L1. Is it a possibility that the treatment with an anti-PD-L1 therapeutic antibody reduces the expression of PD-L1 or blocks the detection of PD-L1, by the IHC detection antibody? Adding a statement clarifying it with references can be helpful.
Round 2
Reviewer 2 Report
Dear Authors
Thank you for response and available data about patients with PR, but you did not provide additional data and separate analysis about patients with SD.